# Effect of Short Fibres in the Mechanical Properties of Geopolymer Mortar Containing Oil-Contaminated Sand

**DOI:** 10.3390/polym13173008

**Published:** 2021-09-05

**Authors:** Rajab Abousnina, Haifa Ibrahim Alsalmi, Allan Manalo, Rochstad Lim Allister, Omar Alajarmeh, Wahid Ferdous, Khouloud Jlassi

**Affiliations:** 1School of Engineering, Faculty of Science and Engineering, Macquarie University, Sydney, NSW 2109, Australia; 2Centre for Future Materials (CFM), University of Southern Queensland, Toowoomba, QLD 4350, Australia; Haifaalsalmi@hotmail.com (H.I.A.); Allan.Manalo@usq.edu.au (A.M.); Omar.Alajarmeh@usq.edu.au (O.A.); Wahid.Ferdous@usq.edu.au (W.F.); 3TuffChem Environmental Services Pte. Ltd., Singapore 508960, Singapore; allister@tuffchem.com; 4Department of Civil Engineering, Tafila Technical University, Tafila 66110, Jordan; 5Centre for Advanced Materials, Qatar University, Doha 2713, Qatar; khouloud.jlassi@qu.edu.qa

**Keywords:** short fibres, contaminated sand, mechanical properties, geopolymer mortar, fly ash

## Abstract

Sand contaminated with crude oil is becoming a major environmental issue around the world, while at the same time, fly ash generated by coal-fired power stations is having a detrimental effect on the environment. Previous studies showed that combining these two waste materials can result in an environmentally sustainable geopolymer concrete. Incorporating sand contaminated with crude oil up to a certain level (4% by weight) can improve the mechanical properties of the produced geopolymer concrete but beyond this level can have a detrimental effect on its compressive strength. To overcome this challenge, this study introduces short fibres to enhance the mechanical properties of geopolymer mortar containing fine sand contaminated with 6% by weight of light crude oil. Four types of short fibres, consisting of twisted polypropylene (PP) fibres, straight PP fibres, short glass fibres and steel fibres in different dosages (0.1, 0.2, 0.3, 0.4 and 0.5% by volume of geopolymer mortar) are considered. The optimum strength was obtained when straight PP fibres were used wherein increases of up to 39% and 74% of the compressive and tensile strength, respectively, of the geopolymer mortar were achieved. Moreover, a fibre dosage of 0.5% provided the highest enhancement in the mechanical properties of the geopolymer mortar with 6% crude oil contamination. This result indicates that the reduction in strength of geopolymer due to the addition of sand with 6% crude oil contamination can be regained by using short fibres, making this new material from wastes suitable for building and construction applications.

## 1. Introduction

Crude oil is one of the most important sources of energy and it plays a great role in improving the economy of any country. However, as shown in Figure 1, oil spills during oil extraction and production are becoming a major environmental concern worldwide [1]. For example, significant amounts of oily wastewater are generated during oil production, which contaminate the surrounding sand [2,3,4]. Moreover, crude oil contamination has a direct effect on the erosion of sand and water infiltration and may cause fire on the ground [5]. Crude oil contamination also affects the physio-chemical characteristics of sand [6]. Sharma and Reddy [7] concluded that the intrinsic permeability (*k*) of contaminated sand increases when there is an increase in density and a decrease in the viscosity of the fluid filling the voids. When the permeability of the sand increased due to the decreased viscosity of the crude oil, the crude oil spread faster, and hence a larger area was affected. Furthermore, the possibility of the crude oil contamination reaching the underground water was higher. Several studies [8,9] have indicated that ground water contaminated by crude oil and other petroleum-based liquids is becoming a widespread problem. 

Several remediation methods have been implemented in order to minimise the adverse effects of crude oil contamination on the environment. However, most of these current remediation approaches are not cost effective [10]. In every instance, clean-up requires knowledge of the mechanical properties of the contaminated sand and the level of crude oil contamination to determine the appropriate remediation method in terms of cost and efficiency. Eagle, et al. [11] indicated that soil washing presents the cheapest remediation method in terms of capital cost, but this method requires up to 23 months. Other methods include vitrification, biological oxidation, chemical oxidation, soil stabilization and encapsulation with cement. These methods are also both time consuming and costly. When encapsulating with cement after lime stabilization, toxic encapsulation was seen in many cases not permanent, and leaching was observed after a period. Encapsulating toxic waste with alkali binders forming geopolymers concrete, mortar and blocks is well documented and has a good track record. 

An alternative and cost-effective method being explored by a number of researchers is to mix oil-contaminated sand in cement and concrete production. The end product can then be used in different engineering applications [12]. Ajagbe, et al. [3] investigated the effect of crude oil on the compressive strength of concrete. They concluded that 18% to 90% of the compressive strength was lost due to 2.5% to 25% crude oil contamination. In another study, oil solidification using the direct immobilization method [13] was implemented to produce cement mortar utilising oil-contaminated sand. Some studies have been carried out to determine the beneficial use of contaminated sand in construction. Furthermore, several researchers have extensively investigated the effects of different percentages of light crude oil contamination (0.5, 1, 2, 4, 6, 8, 10 and 20% by weight) on the mechanical properties and microstructure of produced concrete [14,15,16]. These studies revealed concrete with light crude oil contamination can retain most of its compressive and splitting tensile strength at a contamination level of up to 6%. A good bond between the steel reinforcement and concrete can also be achieved up to this level of oil contamination. Recent studies conducted on geopolymer cement mortar containing fine sand contaminated with light crude oil show that geopolymer mortar has the potential of utilising oil-contaminated sand and reducing its environmental impacts [12,14]. These studies have shown that utilising sand with oil contamination up to a certain level can enhance the properties of the produced concrete.

A significant amount of research and developments on utilising fly ash in geopolymer concrete production have been implemented. Geopolymer is the result of reactive material that is rich in silica and alumina, with alkaline liquid. This material shows promise as a greener substitute for Ordinary Portland Cement (OPC) in some applications. This conclusion was derived due to the geopolymer concrete having good engineering properties and utilising waste materials in its production [17]. Geopolymer concrete is non-flammable and non-combustible and has a better heat and fire resistance than OPC concrete [18]. Geopolymer concrete materials offer economic benefits, as it is estimated that it is 10 to 30% cheaper than OPC, due to the lower cost of fly ash compared to the same weight of cement. Other benefits include carbon credit trading because the appropriate use of one tonne of fly ash creates approximately one carbon credit, with a redemption value of 10 to 20 Euros. In order to provide a more cost-effective solution, oil-contaminated sand can be combined with a cement binder that comes from industrial waste like geopolymer concrete. Recently Abousnina, et al. [12] investigated the suitability of producing geopolymer cement mortar using oil-contaminated sand. The geopolymer mortar with 1% light crude oil contamination yielded 20% higher compressive strength than OPC mortar containing sand with a saturated surface dry condition. However, increasing the crude oil content to 6% caused a significant reduction in compressive strength due to the saturation of the sand particles with oil. New options should therefore be explored to effectively utilise the combination of these waste materials in concrete production suitable for building and construction.

Using fibres as reinforcements in concrete is not new, as they have been used since ancient times [19]. Steel, glass-reinforced concrete (GRC) and synthetic fibres such as polypropylene fibres are now popularly used in enhancing properties of concrete [20]. Steel fibres are known to improve the tensile and flexural strengths of concrete due to their high energy absorption and ability to control crack propagation [21,22,23]. Glass fibre on the other hand has an excellent strengthening effect [24,25], and synthetic fibre has environmental sustainability [26]. Synthetic fibres can prevent the formation of plastic shrinkage cracks in fresh concrete and also improve its post-cracking behaviour [27,28]. Kumar and Naik [29] introduced 0.05%, 0.1% and 0.15% of fibres by volume of cement. Their tests showed a 40% reduction in the drying shrinkage of multi-filament and fibrillated fibrous concrete compared to normal concrete; this reduction is due to the high tensile strength of synthetic fibres contributed to carry more stresses [30]. Fibrillated fibre was also found to control the drying shrinkage better than multi-filament fibre and is recommended for constructing pavements. Cominoli, et al. [31] found that the concrete cover can be reduced resulting in a lighter weight panel and lower transportation costs compared to a normal precast panel. Furthermore, Zeng, et al. [32] investigated the hybrid fibre-reinforced cementitious composites (HFRCCs) using steel fibres, polyvinyl alcohol (PVA) fibres and calcium carbonate whisker (CW) with high temperature resistance and cost-effectiveness. They have concluded that partially replacing PVA fibres by CW could reduce the deterioration of strengths and flexural toughness. Moreover, the calcium carbonate whisker (CW) was used as a cost-effective and environment friendly microfibre in reinforcing cementitious composites [33]. In this study, the influence of high temperature on the micro-structure of CW-reinforced cement paste by nanoindentation and mercury intrusion porosimeter test was investigated. It was concluded that the fractal dimension of the CW-reinforced cement paste was increased with the increased temperature and porosity. 

The above studies highlighted the effect of different fibre lengths and ratios for specific physical and mechanical properties of concrete. However, how these fibres affect the mechanical properties of geopolymer mortars containing light crude are still unknown. This study therefore evaluated the properties of geopolymer mortar containing high percentages of crude oil contamination and with the addition of short fibres. Four different fibres at five different dosages were added to geopolymer mortar containing oil-contaminated sand (6%) and their mechanical and physical properties were evaluated and analysed. The result of this study is anticipated to provide new materials utilising environmentally problematic wastes but with properties suitable for building and construction application. 

## 2. Materials 

### 2.1. Fine Aggregate

The fine aggregates was used because of its similarity to the sand in the Libyan Desert where the first author originated [34]. Fine sand was air dried to meet the condition of fine Libyan Desert sand with a Particle Size Distribution (PSD) shown in Figure 2a determined in accordance with AS 1141.11.1-2009 [35]. Figure 2b shows the contaminated sand on site. The particle grading curve of fine sand shows that these particles are less than 2.36 mm.

### 2.2. Light Crude Oil

Mineral Fork w2.5 motorcycle oil was used as light crude oil because its density and viscosity are similar to the light crude oil [36,37] that commonly contaminates the sand in Libya. A comparison of the important properties of this oil is shown in Table 1.

### 2.3. Fly Ash

The fly ash used in this study was Type F (low calcium) fly ash of approximately 15 µm. It was sourced from Pozzolanic Millmerran, Queensland, Australia. The chemical composition of the fly ash is given in Table 2, and its packing density is 1100 kg/m^3^.

### 2.4. Alkaline Liquid 

A combination of sodium silicate (Na_2_SiO_3_) and sodium hydroxide (NaOH) solutions were used as alkaline liquid. The sodium silicate solution was obtained from PQ, Sydney, NSW, Australia. This solution is recommended for use as an ingredient for detergent, adhesive, binder, the source of feedstock silica or industrial raw material; the properties of sodium silicate solution are presented elsewhere [12]. The sodium hydroxide solution was prepared in the laboratory by dissolving pellets of sodium hydroxide in water. Its specific gravity depends on its concentration and is expressed by the term molar (M). The concentration needed to make geopolymer concrete varies from 8 M to 16 M; in this study, a 10 M solution was prepared. This molarity was selected based on initial trials comparing 10 M and 12 M, where 30% higher compressive strength was achieved with 10 M than with 12 M [12].

### 2.5. Short Fibres

Four short fibres (twisted PP fibre, straight PP fibre, short glass fibre and steel fibre, as shown in Figure 3 and at five different dosages (0.1, 0.2, 0.3, 0.4 and 0.5% by volume)) were mixed with geopolymer mortar containing sand with 6% of crude oil contamination. In the first stage, 0.1% of each fibre was introduced to the geopolymer mortar and mixed for at least 2 min to ensure the fibres were distributed evenly in the sample. The sample was then poured in the mould, compacted, and cured in an oven for 24 h at 60 °C, and then left in an ambient condition for up to 28 days. In the second stage, five different amounts of fibre were applied: 0.1%, 0.2%, 0.3%, 0.4% and 0.5% by volume of mortar. All the samples were then tested after 28 days of curing.

Twisted PP fibre is a fibre twisted into a bundle, which helps the fibres mix into the concrete and be distributed throughout the concrete matrix. This fibre is designed to provide a strong resistance to impact force and is designed to retain its cross-sectional shape. Straight PP fibre is a high-performing polypropylene fibre with a surface uniquely designed to provide anchorage. This design provides excellent resistance as well as improving the toughness and crack control of the concrete. On the other hand, short glass fibre made of 100% virgin homopolymer fibrous reinforcement in a collated fibrillated (network) form is used to reduce the plastic and hardened concrete shrinkage, increase resistance to fatigue and improve the impact strength and concrete toughness. Steel fibre is used to reinforce concrete and control cracking in any situation. It is designed to guarantee stability and improve the rupture forces. The properties of these fibres are provided in Table 3. All the used fibres in this study are commercially available and are used extensively. It is worth mentioning that the PP fibres are waste plastic products, which provides more environmental as well as cost benefits.

### 2.6. Mix Design

The materials used in the mortar mix are shown in Figure 4. The mortar was mixed manually in the laboratory using a Kitchenaid mini mixer (mono phase 5lt). Samples were prepared by mixing dry sand with 6% of light crude oil by weight. The oil was mixed with the dry sand and then the samples were placed inside a plastic container for 72 h to allow the mixture to become homogenous. A lid was placed on the plastic container to prevent the crude oil from evaporating from the sand. Fly ash was then added to the fine sand and mixed for about two minutes. Alkaline solutions were prepared by mixing sodium hydroxide (NaOH) and sodium silicate (Na_2_SiO_3_) solutions at least one day before mixing the geopolymer mortar; the solutions were gradually added to the solids and mixed for another two minutes. Fibres were then added to the mixture and mixed thoroughly. The mix design of this geopolymer mortar was in accordance with AS 2350.12-2006 [38] having mix proportions of 1:3:0.5, that is, one part fly ash and three parts sand at a fixed ratio of fly ash to alkaline solution of 0.5. This mix is comparable to a previous study on cement and geopolymer mortar that contained fine sand contaminated with light crude oil [12]. Plastic moulds (50 mm in diameter by 100 mm high) were used to avoid using any releasing agent or grease to remove the specimens. This also prevented any crude oil leaching from the mix. All the samples were kept in an oven at 60 °C for 24 h, and then the specimens were stored in a fog room at 25 °C and RH = 85%. The temperature and RH were monitored using a digital thermometer and humidity meter.

#### 2.6.1. Property Characterisation of Mortar with Different Short Fibres

The effect of short fibres on the physical and mechanical properties of geopolymer mortar containing 6% of crude oil contamination has been investigated. In this stage, fly ash was mixed with the alkaline solution, sand was then added, followed by fibres. Only one type of fibre was used for each stage, at a concentration of 0.1% fibre. Each fibre sample (three specimens produced for each fibre) was then tested under compressive and splitting tensile loading. The test outlines are tabulated in Table 4.

#### 2.6.2. Properties with Different Dosages of Short Fibres 

Fly ash was mixed with the alkaline solution. This was followed by adding sand and different amounts of fibre (the type of fibre selected is based on the optimum results of stage 1). The samples were then tested under compressive and splitting tensile. The dosages of fibres in this stage were 0.1%, 0.2%, 0.3%, 0.4% and 0.5% by volume of mortar; they were implemented to enable a comparative study to be undertaken, and the test outlines are tabulated in Table 5.

### 2.7. Compressive Strength Test of Geopolymer Mortar

The cylindrical specimens were surface ground before being placed in the testing machine to make the top and bottom surfaces smoother to ensure a more uniform load distribution, shape and dimensions. The compressive strength of three samples of each mix was tested and the average of each batch was plotted. Three cylinders, 50 mm×100 mm in size were tested from every mix; this represents a crude oil contamination of 6% with different types and percentages of fibres. Testing was carried out in accordance with AS 1012 (1999). A load was applied using a 100 kN MTS machine at a constant cross head speed of 1 mm/min, as shown in Figure 5. The compressive strength was calculated by dividing the load by the cross-sectional area of the specimens. The failure mechanisms of each specimen were also observed and recorded. 

The interior part of the specimen sections was analysed (Figure 6) under a microscope set (Motic SMZ-168 series) at a magnification of 65 times. The test was carried out after completing a compressive strength test after 28 days of curing.

## 3. Results and Observations

### 3.1. Effect of Different Fibre Types

#### 3.1.1. Compressive Strength

Generally, adding fibres to the control sample has showed similar modes of failure in terms of initiation and progression. It has been observed that all specimens with fibres (short glass fibres, steel fibres, twisted PP fibres and straight PP fibres) displayed signs of shear failure. However, the results of the compressive stress revealed that the presence of fibres in geopolymer materials affects the compressive strength, as shown in Figure 7. The average strength of the control (non-fibre with 6% of crude oil contamination) specimens was 11.63 MPa while the specimens with fibres (short glass fibres, steel fibres, twisted PP fibres and straight PP fibres) increased in strength by 4%, 20%, 33% and 39%, respectively. Moreover, the specimens with fibres (short glass fibres, steel fibres, twisted PP fibres and straight PP fibres) increased in axial toughness by 47%, 77%, 102% and 137%, respectively, compared to the control sample (see Table 6). The highest enhancement achieved from adding straight PP fibres was due to the external deformation on the surface of the fibres, which promoted a bond with geopolymer and helped absorb the maximum energy during compression. Furthermore, the surface geometry of the twisted PP fibres was able to record higher compressive strength and toughness compared to the plain steel fibres owing to the increase in the bond to the concrete. On the other hand, the shape and texture of the short glass fibres did not effectively enhance its ability to bond with the geopolymer mortar, possibly because the oil in the mortar impede the fibre’s ability to anchor itself to the mortar as well as its shortness in length compared to the other fibre counterparts. 

Investigating the stress displacement between these specimens helped clarify the dynamic relationship between the fibres in the mortar and the geopolymer and oil contaminated sand. The comparison curve of stress displacement for the non-fibre specimen with 6% of crude oil contamination and the specimens with four different types of fibres revealed that the straight PP fibre specimens displayed a softening peak as they reached their maximum strength at greatest displacement, unlike the control specimen, which had a lower load and displacement. The steel, twisted PP fibres and short glass fibres reinforced mortar experienced an increased post-peak behaviour. Several researchers [39,40,41,42] showed that the presence of fibres modified concrete specimens’ failure behaviour by enhancing its mechanical properties and reducing the material’s brittleness. The behaviour of the mortar in this study reflects the research outcomes of Al-Majidi, et al. [43] where they concluded that the addition of fibres to mortar increased its compressive strength. These results also showed that peak stress for the twisted PP fibres, short glass fibres and steel fibres were reached at increased displacements, which concurs with investigations by Holcim Australia [44] and Maccaferri [45] who stated that adding additional fibres to the concrete enhanced the matrix and enabled it to carry greater tensile loads. The fibres, distributed homogeneously, improve the tensile strength and create a micro-scaffolding that leads to more supple concrete and also helps control the formation of cracks due to shrinkage. Recently, Yu, et al. [46] investigated the physical, mechanical and microstructural properties of epoxy polymer matrix with crumb rubber and short fibres for composite railway sleepers. It has been concluded that up to 10% of weight of GFCS has been observed successfully, enhancing all mechanical properties of the PFR mix. 

#### 3.1.2. Splitting Tensile Behaviour

The splitting tensile strength of geopolymer mortar containing oil contaminated sand is summarised in Table 7 and Figure 8a. All specimens containing fibres increased their splitting tensile strength. For example, the straight PP, twisted PP, short glass, and steel fibres specimens were greater than the control specimen by 74%, 60%, 63%, and 63%, respectively. These results reflect the work of Shaikh [47] who concluded that adding short fibres to reinforced geopolymer composites significantly improved their flexural and tensile strengths. Fibres help the mortar to bond, which then strengthens the mortar while increasing its resistance to absorption of loads. The investigation by King [48] found that straight PP fibres significantly improved the compressive and flexural strength of concrete containing crude oil contamination due to their strong bond to mortar because of its tread surface anchors.

Figure 8b shows the curve of the relationship between the fibre and control sample (non-fibre specimens). It shows that all the specimens with fibres have a softening curve at its peak as they reach their maximum load. It can be seen that all the specimens showed almost similar stiffness at the initial loading level (where the stiffness is normally taken before the crack initiation), which refers to the relatively small number of fibres compared to concrete. In this case, the stiffness of concrete will be dominant. On the other hand, the effect of the fibres’ stiffness can be noticed after observing the crack where the specimens with steel fibres showed the best performance in retaining the strength afterwards. The load and displacement were higher than the non-fibre sample. Straight PP fibres showed an increased failure load and displacement, while the control specimen produced a failure load and displacement in less time. Fibres modified the behaviour of geopolymer mortar failure, but this modification increased the ductility unlike in the non-fibre specimens. The results of the study by Kooiman and Walraven [42] indicated that the change in strength stems from a softening response caused as the stresses of the fibre bridging are re-distributed to create a new state of equilibrium across the cross-section after cracking; this potentially results in the maximum moment capacity exceeding the maximum moment of plain concrete. On the other hand, it can be observed (Figure 8b) that all the specimens showed almost similar stiffness at the initial loading level (where the stiffness is normally taken before the crack initiation), which refers to the relatively small number of fibres compared to concrete. In this case, the stiffness of concrete will be dominant. On the other hand, the effect of the fibres’ stiffness can be noticed after observing the crack where the specimens with steel fibres showed the best in retaining the strength afterwards. Adding fibres increased the tensile strength of the mortar, for example, increases of up to 60%, 63% and 74% were achieved by adding Twisted PP, short glass, Steel and Straight PP fibres, respectively, compared to the control specimen. Straight PP fibres outperformed all other fibre specimens, presumably because its tread surface design acted like an anchor to enhance mortar bonding. This fibre played a critical role in improving the mortar because of its ability to provide a complex micro-crack matrix to prevent crack bridging. All the fibre types increased ductility by displaying increased displacements at failure loads. The non-fibre specimen had a lower displacement at low loads compared with the fibre specimens because fibre can redistribute the tensile stresses and thus enable the fibre-reinforced specimens to not fail suddenly. Straight PP fibre had the highest tensile strength and improved the geopolymer mortar ductility of all the fibres. This type of fibre was therefore used for the next stage where the optimum dosages that will further enhance the physical and mechanical properties of geopolymer mortar with oil-contaminated sand were investigated. 

At a microscopic level, as seen in Figure 9, the straight PP fibres, Twisted PP fibres and steel fibres all revealed a strong bond to the mortar, clearly indicating that their shape contributed to the fibre–matrix interface. Straight PP fibre bonded strongly with the mortar. Its tread surface design helps anchor it to the mortar bond while also absorbing maximum energy during compression. Twisted PP fibres formed strong bonds with the mortar because of its twisted bundle nature. The fibre broke up into separate filaments after mixing and these structures intertwined with the mortar. The steel fibres also formed a solid bond with the mortar, and this was enhanced by the fibre’s bent ends. Furthermore, Figure 9a,c,d shows that there are geopolymer mortar formations along the length of fibre. Moreover, there are no visible air voids between the fibres on the mortar. However, microscopic images of the short glass fibre showed that the voids surrounding are smooth, soft fibres. It is assumed that the presence of oil in the mortar impeded these smooth strands from anchoring to the mortar.

### 3.2. Effect of Fibre Dosages 

#### 3.2.1. Compressive Strength

All specimens with 1%, 0.2%, 0.3%, 0.4% and 0.5% of straight PP fibres demonstrated increased compressive strength ($\sigma_{c}$) compared to the control non-fibre geopolymer mortar. The strength increase is directly related to the increased volume of fibre. The average compressive strength for each group of fibre dosed ranged from 0.1% to 0.5%, and were calculated as 19.1 MPa, 24.4 MPa, 25.0 MPa, 31.1 MPa and 32.2 MPa, respectively. There was a minor standard deviation in strength (0.15) of the 0.1% and 0.5% doses of fibre, where the largest deviation was 1.45 for the 0.4% dose due to the extended range between the strength of the first sample (29.4 MPa) and the second sample (32 MPa). The horizontal line in Figure 10a indicates the average compressive strength (11.63 MPa) of the control (non-fibrous with 6% of crude oil contamination) specimens. The optimum strength was observed when 0.4% fibre was used with an increase in compressive strength of 39% more than the strength of the 0.1% fibre dosage. Dosages of 0.2%, 0.3% and 0.5% performed 22%, 24% and 41%, respectively, better than the 0.1% group. These outcomes reinforce the concept that there is a distinct relationship between fibre dosages and compressive strength. Moreover, high consistency in the compressive strength values can be observed in the fibrous concrete samples compared to the control samples. This is evidenced by the low standard deviation and coefficient of variance values, as reported in Table 8 and Table 9.

The findings that increased dosages of fibres are proportionate to the increase in the compressive strength for geopolymer mortar with 6% oil-contaminated sand is in contrast with the results of Neves [49,50], where increased fibre dosages did not substantially increase the compressive strength of specimens. This is, however, in agreement with the results obtained by Lee, et al. [41] and Shaikh [47] where the former found that the compressive strength of SFRC increased with larger doses of hooked steel fibre, and the latter found that adding additional fibres to mortar blended with fly ash increased its compressive strength.

The anomalies raised by Neves [49,50] may be due to other variations in their research, such as the materials and their parameters. For example, both research projects experimented with concrete, rather than mortar, thus requiring different material ratios. Neves [49] used amorphous metallic fibres instead of straight PP fibres. On the other hand, Braden [50] used steel fibres with concrete containing 10% of crude oil contamination. In this study, the author noted that the use of fibres in the concrete’s compressive strength was a delicate balance between micro-crack bridging and additional voids caused by the addition of fibres. When the stress–displacement behaviour of geopolymer with 6% oil-contaminated sand was analysed, all five specimens became softer after reaching their maximum load, as they continued towards their highest displacement. The 0.1% fibre group displayed the softest curve and lowest registered load while the 0.5% group demonstrated the sharpest peak. Both the 0.4% and 0.5% fibre dosage groups demonstrated the highest loads, but there was less displacement by the 0.5% group than the 0.4% group, indicating that the latter group increased its post-peak behaviour at the lowest displacement. In 2015, Lee, et al. [41] found that SFRC demonstrated ductile behaviour after reaching a compressive failure load, and the strain at this load generally increased when the fibre ratio and aspect ratios were also increased. These findings support the results of this study, where an increased dose of straight PP fibres increased the compressive strength of geopolymer with 6% oil-contaminated sand. Recently, Yu, et al. [46] investigated the effects of increasing the content of crumb rubber, chopped glass fibres and polypropylene fibres on the physical, mechanical and microstructural properties of the epoxy polymer matrix. Their results showed that up to 10% weight of GFCS has been observed to successfully enhance all mechanical properties of the PFR mix. However, increasing the fibre content caused unacceptable workability and a high porosity, which resulted in a sudden deterioration in compressive and shear strength. Furthermore, a 0–1.5% volume ratio of macro-Polypropylene (PP) fibres had a slight effect on the flow and density of the PFR samples but gradually increased the porosity. 

In this study, increasing the fibre content up to a certain amount did not affect the workability due to the presence of 6% of crude oil. This agrees with the previous studies that indicated that crude oil works as a plasticiser agent, thus as the amount of crude oil increased, the workability increased as well. Hamad and Rteil [51] stated that the oil acted like a chemical plasticiser and improved the fluidity and doubled the slump of the concrete mix, while maintaining its compressive strength. Furthermore, the increase in flow for mixes containing light crude oil is a similar outcome of other studies [51,52]. In this study, the presence of 6% of crude oil played a great role in decreasing the effect of workability as the amount of fibre increased. There was no significant effect on the workability up to 0.3 fibres; however, as the amount of fibre increased to 0.4 and 0.5, the workability decreased silently. By increasing the amount to 0.6, the workability of the geopolymer mortar was very low so this % was excluded. 

Displacement at failure did not increase with the addition of more fibres—the 0.1% fibre specimen registered 2.4 mm displacement while the specimen with 0.5% fibre measured 2.2 mm. These failures indicate that an increased volume of fibre does not affect the types of cracks because most specimens demonstrated shear and splitting failure, but an increased dose of fibre reduced the number of cracks present. Therefore, the specimens containing the larger ratios of fibre have a greater probability of increasing the bond energy and resisting cracking and crack propagation.

#### 3.2.2. Splitting Tensile Behaviour

The tensile strength of mortar reinforced with straight PP fibres increased as the dose of fibre increased, with 0.5% recording higher strength than all the other fibre ratios. The difference in splitting tensile strength between the 0.5% dosage and 0.1% dosage is 89%. The standard deviation of the 0.5% fibre ratio was higher because the data points were spread over a wider range of values, but the standard deviation of the 0.1% fibre dosage remained the lowest, indicating that the data points were quite close. The average strength of each fibre dosage of 0.2%, 0.3%, 0.4% through to 0.5% was 2.9 MPa, 2.8 MPa, 4.1 MPa and 5.4 MPa, respectively.

The effect of doses of straight PP fibre on the splitting tensile strength is summarised in Figure 10b. A red line is drawn at 0.6 MPa, representing the average tensile strength of non-fibre specimens with 6% of crude oil contamination. Specimens with 0.2%, 0.3% and 0.4% of fibre showed an increase in strength of 79%, 79% and 86%, respectively. The optimal strength was observed when 0.5% fibre was used, and it showed an improvement of 89% compared to the control (non-fibre specimens with 6% crude oil contamination). These results show a significant difference between the 0.1% and 0.5% dosage rates, indicating that an increased dose of fibre significantly increased the tensile strength of geopolymer with 6% of crude oil contamination. A similar result occurred when Shaikh [47] added fibre to a fly ash mixture, as it significantly improved the tensile strength at early stages up to 28 days. King [48] also found that an increased dose of Straight PP fibre enhanced the flexural strength of concrete with 10% of oil contaminated sand. These results indicated that the increased doses of fibre improved crack bridging and its load transfer. This bridging stemmed the movement and size of cracks at the point of failure, as well as improving the tensile capacity of mortar. Furthermore, when analysing the load–displacement behaviour, the introduction of Straight PP fibres distinctly improved the strength and displacement, so the specimen with 0.5% fibre experienced an increasing failure load and displacement of 5.4 MPa and 1.7 mm, while the 0.1% specimen registered at 2.3 MPa and 0.7 mm. This indicates that an increased dose of fibre increased the failure load and at a greater displacement. Furthermore, the fracture surface of the specimen with and without fibres was imaged under a microscope. The stereo microscope image in Figure 11 shows the ability of longitudinal fibres to control internal cracking resulted in enhancing the structural properties. In this order, this will lead to increasing the durability of concrete in general as the fibres are able to block the internal crack initiation and propagation. 

## 4. Theoretical Predictions of the Compressive and Tensile Strength 

It is well known that the tensile capacity of the concrete represents about 10% of its compressive strength due to its brittle behaviour [53]. The inclusion of reinforcements, however, significantly improves the tensile resistance of the reinforced concrete elements. In this study, the addition of 6% oil-contaminated sand significantly affected the mechanical properties of the concrete wherein its tensile strength is only 5.6% of its compressive strength (see Table 10). This is due to the separation between the sand particles and cement caused by the oil contamination, which eases the crack initiation [12]. The addition of short fibres, however, significantly improves both the tensile and compressive strength of the fibrous geopolymer mortar. Based on the measured strength enhancement, this study proposes empirical formulas to theoretically predict the mechanical properties of the PP fibre-reinforced geopolymer concrete with oil-contaminated sand. In Figure 12a, the increase in the compressive strength was dominated by the fibre dosage at which the use of 0.5% revealed the highest compressive strength enhancement by 277% compared to the control samples (without fibres). Therefore, Equation (1) is suggested to predict the compressive strength of the concrete with respect to the fibre dosage addition, which shows good agreement with the experimental results (see Table 10). Moreover, the experimental tensile strength results were compared to the ACI code recommendations (Equation (2)) [54], which showed high variation attributing to the use of contaminated sand in the concrete mix, as seen in Figure 12b. It is worth mentioning that Equation (2) significantly overestimates the tensile strength of the control samples and the ones with a low amount of fibre dosage. However, it also underestimates the tensile strength of the samples with high fibre dosage. This confirms the need to develop a new empirical formula, which can predict the tensile strength of the concrete with oil-contaminated sand and various PP-fibres. Therefore, a new expression was suggested to predict the tensile strength of the tested samples (see Equation (3)). Figure 12c and Table 10 show good agreement between the theoretical and experimental results. Moreover, Figure 12c presents the match between the theoretical and experimental results for both compressive and tensile strength results in this study, which reveals good agreement. However, the predicted tensile strength of the control samples shows high variation from the experimental one. This is due to its low tensile/compressive strength ratio where Equation (3) used the theoretical compressive strength resulted from Equation (1). The following equations, $f_{c}^{\prime }$, $f_{c-th}^{\prime }$, $f_{t-ACI}$, $f_{t-th}$ and α represent the experimental compressive strength, theoretical compressive strength, theoretical tensile strength relying on ACI code (Equation (2)), theoretical tensile strength (Equation (3)) and fibre dosage, respectively.
(1)$$f_{c-th}^{\prime }\left(\alpha \right)={\left(f_{c}^{\prime }\right)}^{\left(1+\sqrt[{3}]{15\alpha }\right)}$$
(2)$$f_{t-ACI}\left(\alpha \right)=0.62\sqrt{f_{c-th}^{\prime }}$$
(3)$$f_{t-th}\left(\alpha \right)=k\sqrt{f_{c-th}^{\prime }}=\frac{1}{3}e^{{\left(100\alpha \right)}^{2}}.\sqrt{f_{c-th}^{\prime }}$$

## 5. Discussion

It is well known that the steel-reinforcing bars help the structure withstand the tensile stress acting on it by providing more ductility and strength. However, the corrosion of steel reinforcement is a huge concern in the construction industry because in many instances contractors are unable to maintain the minimum cover needed to inhibit further corrosion [55,56]. Therefore, steel, glass, natural, and synthetic fibres have started to be considered as an alternative reinforcement [57]. Synthetic fibres can prevent the formation of plastic shrinkage cracks in fresh concrete and improve its post-cracking behaviour [27,28]. Since normal concrete is brittle, a concrete structure can only withstand low tensile strain and strength, which is why more ductile synthetic fibres are increasingly being used in general practice. Shah, et al. [58] states that since concrete is likely to have micro cracks during its initial stages, synthetic fibres are used to improve the quality of normal concrete. For instance, glass fibre has an excellent strengthening effect [24,25], and synthetic fibre has environmental sustainability [26]. Recently, Yu, et al. [46] investigated the effect of different mixes of the crumb rubber and short fibre reinforcement on the physical, mechanical properties and microstructure of the epoxy-based PFR system. Results showed that short fibres enhanced the flexural and shear performance while the crumb rubber improved flexibility of polymer mixes. As shown above, using syndetic fibre has already been considered and it has improved the properties of concrete. However, it has been reported that as the amount of fibre increased, the workability decreased and consequently negatively affected the properties due to the increase in the porosity. On the other hand, the use of oil-contaminated sand in construction is now being considered as an alternative and cost-effective remediation method to minimise its adverse effect in the environment. Furthermore, Abousnina, et al. [12] investigated the effects of light crude oil contamination on the physical and mechanical properties of geopolymer cement mortar. Results showed that the geopolymer mortar with 1% of light crude oil contamination yielded a 20% higher compressive strength than OPC mortar containing sand with a saturated surface dry condition. Nevertheless, increasing the crude oil content to 6%, 8% and 10% caused a significant reduction in compressive strength (26.4, 36.7 and 40.7%, respectively) due to the saturation of the sand particles with oil. Conversely, increasing the crude oil content increases the workability, which indicates that the crude oil works as a plasticiser agent [59]. This study therefore evaluated the properties of geopolymer concrete containing high percentages of crude oil contamination (6%) and with the addition of different short fibres. Since the main purpose of this study was to understand the behaviour of fibre-reinforced geopolymer mortar with the addition of 6% of oil contaminated sand, the fibre length, which ranged 19–45 mm, did not affect the general trend objectives of the result study as all the samples were treated similarly. The increment of crude oil contamination has shown an improvement of the workability; this will overcome the reduction of the workability as the amount of fibre increases. The introduction of straight PP fibres generally improved the ductility of geopolymer mortar with 6% of oil contamination, with the increased volume of fibre directly influencing the tensile behaviour. The displacement of the specimens with 0.5% dose of fibre at a maximum load was 1.8 mm with 5.4 MPa strength, while the displacement of the non-fibre control specimen registered at a lower displacement and load of 0.29 mm and 0.6 MPa strength. Overall, the increased volume of fibre led to an increased load failure at a larger displacement, which indicated that a higher dose of fibre helped to reduce the mortar specimen’s brittle behaviour while increasing its resistance to displacement. The various dosages of short fibres significantly increased the compressive behaviour. 

The addition of varying percentages of fibre improved the softening displacement before and after the ultimate load, and also increased the ductility. Moreover, an increased dose of fibre lengthened the post-peak softening branch and increased the displacement of the specimen at failure load. The specimens with 0.4% fibre displayed the most significant load at a high displacement and extended the softening branch of post-peak mortar. Adding various doses of fibres increased the mortar’s tensile strength of geopolymer mortar with 6% of crude oil contamination. There was a significant increase in tensile strength with all doses of fibre, especially at 0.5%. Furthermore, adding 0.5% fibre by weight to geopolymer mortar with 6% oil-contaminated sand increased its tensile strength by 89%. This increased tensile strength creates a softening response on the graph because the fibre bridging makes the mortar more ductile. This increased ductility can be shown through the failure analysis where there are reduced crack widths and spread as the doses of fibre increased. Moreover, the increased fibre ratio is likely to create a good bond with the mortar and improve the tensile strength.

All the doses of fibre increased the ductility of mortar and displayed increased displacements at failure loads under the splitting tensile test. The geopolymer mortar with 6% oil-contaminated sand with the minimum dose of fibre showed a lower failure displacement than the specimens with higher doses of fibre. These increased doses increased the failure load at a larger displacement. The maximum dose of fibre of 0.5% displayed the highest failure load at a large displacement. The added volume of fibre made the geopolymer mortar with 6% oil-contaminated sand specimen’s fail in ductile behaviour and exhibited higher displacement after reaching their peak strength. In summary, increased dosages of straight PP fibres have a significant effect on the compressive strength of geopolymer mortar with 6% oil-contaminated sand. The stress–displacement behaviour of the mortar with doses of fibre was more ductile while the tensile strength was more positively affected with increased volumes of fibre. The produced geopolymer mortar containing oil-contaminated sand can be used for building and construction. Previous studies [60,61,62] mentioned that concrete with compressive strength varying from 17 MPa to 28 MPa is suitable for residential concrete such as footings, foundation walls and slabs. Even the concrete using fine sand with 10% and 20% oil contamination (compressive strength of 13.87 and 4.76 MPa, respectively) can be used for some low-load-bearing engineering applications such as landfill layering and the production of bricks. According to the United States Environmental Protection Agency (USEPA) guidelines, the recommended compressive strength at 28 days for layering in the landfill disposal site is 0.35 MPa and 1.0 MPa in France and the Netherlands [63], respectively, whereas a higher compressive strength of 3.5 MPa in a sanitary landfill is required by the Wastewater Technology Centre (WTC), Canada [64]. Nevertheless, based on the British standard for precast concrete masonry units (BSI, 1981), a higher compressive strength of 2.8 and 7 MPa, respectively, is required for blocks and bricks, and a minimum of 7 days for a cube compressive strength between 4.5 and 15 MPa is required by the Department of Transport in UK for sub-base and base materials. This shows the high potential of concrete with oil-contaminated sand as a sustainable material in building and construction. In the final analysis, increased doses of fibre can enhance the mechanical properties of geopolymer mortar with 6% oil-contaminated sand.

## 6. Conclusions

This study investigated the effect that adding short fibres has on the physical and mechanical properties of geopolymer mortar containing 6% oil-contaminated sand. Based on the results of this study, the following conclusions can be drawn:Short fibres can significantly improve the compressive strength of geopolymer mortar with the best enhancement achieved using straight PP fibres due to its good bond with the mortar. The compressive strength improved by as much as 39% compared to mortar without fibres.The addition of short fibres regardless of types improved the softening post-peak branch of mortar stress–displacement compared to control specimen (non-fibre).The tensile strength of mortar with 6% crude oil contamination increased with the addition of fibres due to bond between the fibre and mortar, which then strengthens the mortar while increasing its resistance to absorption of loads. Straight PP fibres improved the tensile strength by as much as 74% compared to the non-fibre specimen.All the fibre types improved the ductility of mortar by increasing its displacement at failure loads. This is due to the bond with geopolymer, which helped absorb the maximum energy during compression.The optimum dosage of 0.5% improved the compressive and splitting tensile strength of geopolymer mortar by 65% and 89%, respectively, compared to the non-fibre specimen. Increasing the doses of fibre with the presence of oil helped the homogeneity of fibres and that improved crack bridging and its load transfer.The simplified equations were developed to predict the critical properties of geopolymer with 6% and it shows good agreement between the theoretical and experimental results.

This study showed that the reduction in strength of geopolymer due to the addition of sand with 6% crude oil contamination can be regained by using short fibres, making this new material from wastes suitable for building and construction applications. Thus, further future investigation needs to be conducted on the environmental and durability behaviour of the new geopolymer concrete with 6% of crude oil contamination. The tested fibres had fairly close aspect ratios; the fibre aspect ratio (length-to-diameter ratio) has an impact on the mechanical properties of concrete; thus, it is recommended to be among future research. 

## Figures and Tables

**Figure 1 polymers-13-03008-f001:**
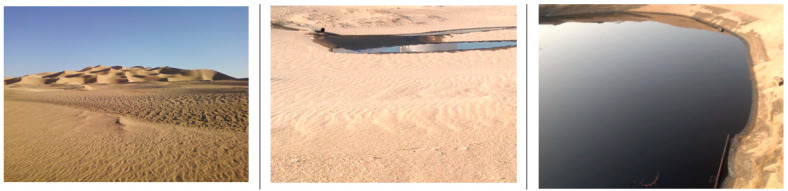
Oil-contaminated sand—Ghani Field–VOO, Libya.

**Figure 2 polymers-13-03008-f002:**
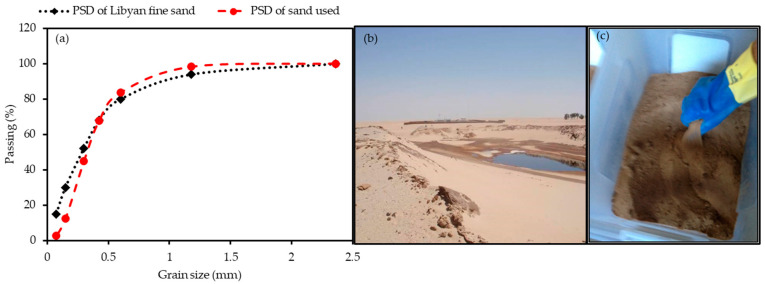
(**a**) Particle size distribution curves of fine sand used and Libyan fine sand [35], (**b**) oil-contaminated sand from the site, (**c**) contaminated sand used in the experiment.

**Figure 3 polymers-13-03008-f003:**
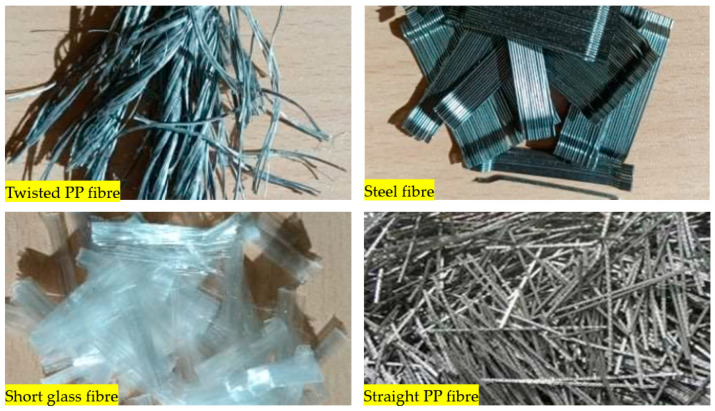
Type of fibres used (twisted PP fibre, straight PP fibre, short glass fibre and steel fibre).

**Figure 4 polymers-13-03008-f004:**
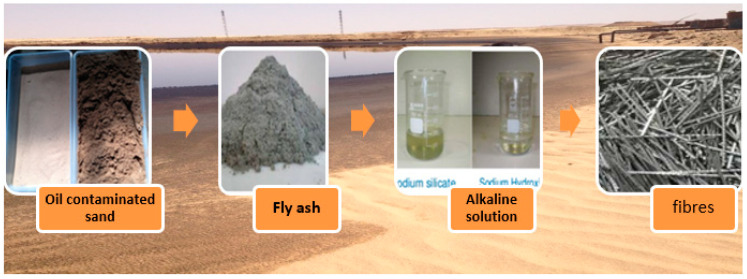
Materials used for mixture of oil contaminated sand, fly ash, alkaline solution and fibres.

**Figure 5 polymers-13-03008-f005:**
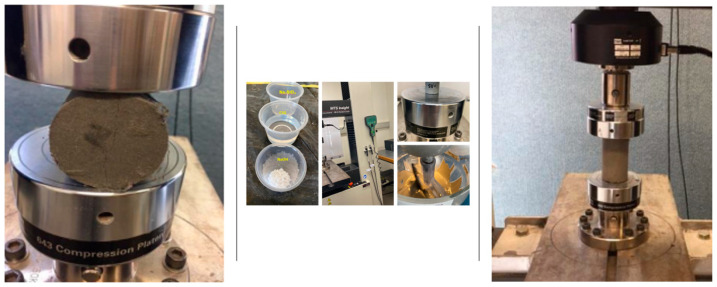
Compression and tensile test set up (MTS 100 kN).

**Figure 6 polymers-13-03008-f006:**
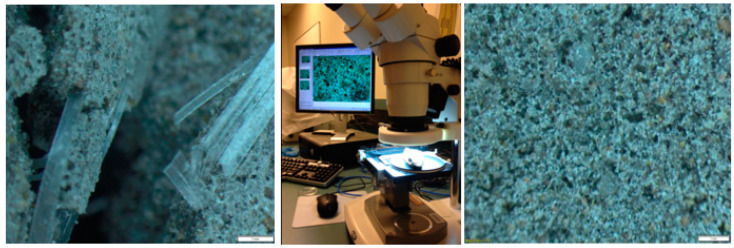
Microstructure observations (Microscope).

**Figure 7 polymers-13-03008-f007:**
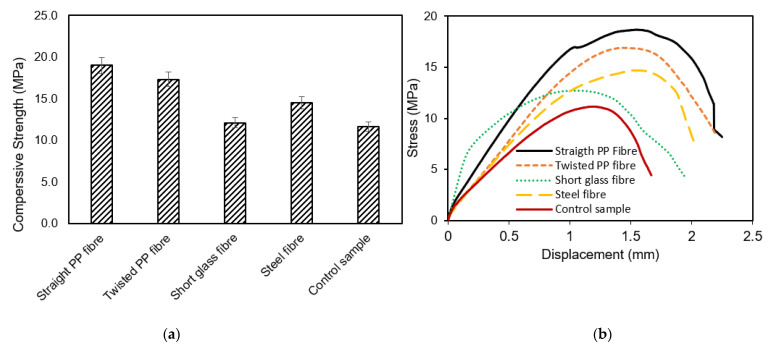
(**a**) The average compressive strength by the type of fibre, and (**b**) typical stress–displacement behaviour of compressive strength with various types of fibres.

**Figure 8 polymers-13-03008-f008:**
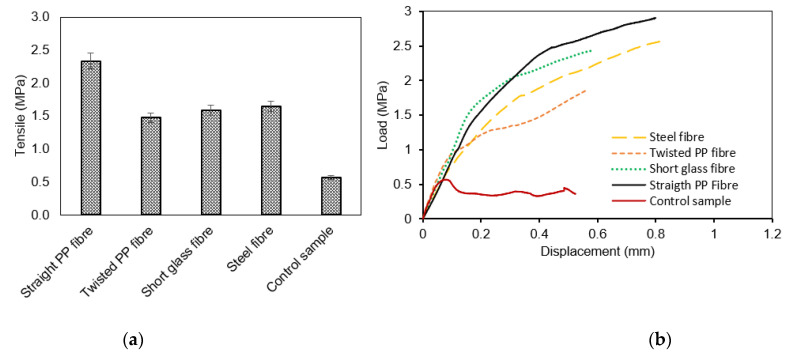
(**a**) Average of splitting tensile strength by fibre type, and (**b**) load–displacement relationship for tested specimens under splitting tensile loading.

**Figure 9 polymers-13-03008-f009:**
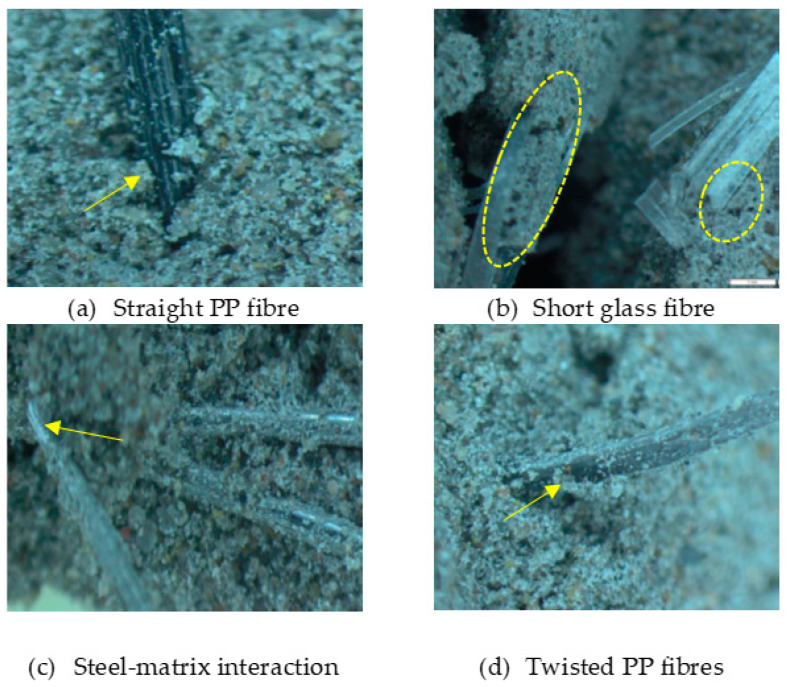
Microscope images of geopolymer mortar with different fibres: (**a**) Straight PP, (**b**) short glass, (**c**) steel fibres and (**d**) twisted PP.

**Figure 10 polymers-13-03008-f010:**
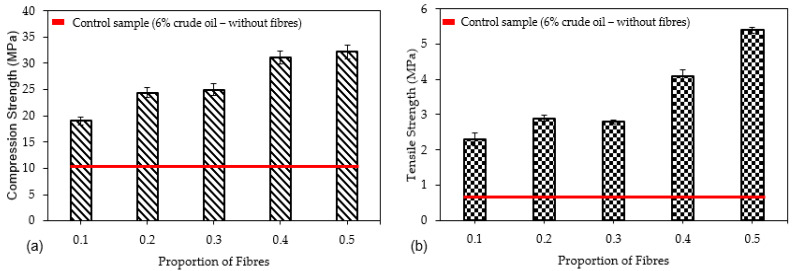
(**a**) Average of compression strength (MPa) and (**b**) Average splitting tensile strength (MPa) with varied straight PP fibres (MPa).

**Figure 11 polymers-13-03008-f011:**
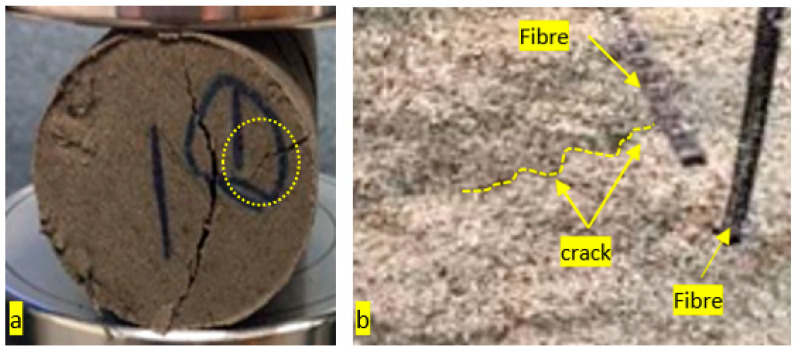
Stereo microscopic image showing the internal cracks.

**Figure 12 polymers-13-03008-f012:**
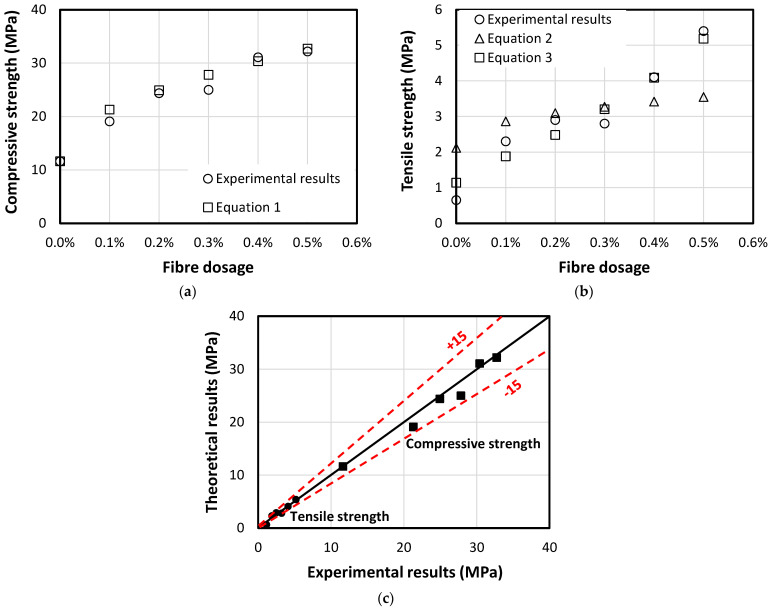
Predictions of mechanical properties of fibrous concrete with oil-contaminated sand, (**a**) compressive strength; (**b**) tensile strength; (**c**) comparison between experimental and theoretical results.

**Table 1 polymers-13-03008-t001:** Comparison between light crude oil and Fork w2.5 Motorcycle oil.

Specifications	Light Crude Oil	Fork w2.5 Motorcycle Oils	Ref.
Density (kg/L)	0.825	0.827	
Viscosity (mm²/s)	5.96	6.74	[36,37]
Temperature (°C)	40	40	

**Table 2 polymers-13-03008-t002:** Chemical composition of fly ash (%).

Element	SiO_2_	Al_2_O_3_	Fe_2_O_3_	CaO	MgO	Na_2_O	K_2_O	SO_3_
Percentage (%)	51.8	24.4	9.62	4.37	1.5	0.34	1.41	0.26

**Table 3 polymers-13-03008-t003:** Properties of fibres used in this study.

	Twisted PP Fibre	Straight PP Fibre	Short Glass Fibre	Steel Fibre
Material	100% virgin co-polymer/polypropylene	100% virgin polypropylene	Virgin homopolymer polypropylene	Bright low carbon steel wire
Form	Twisted bundle non-fibrillated monofilament and a fibrillated network	Monofilament fibre systema	Collated fibrillated fibre	Round wire, hook shape
Fibre Count	161,900/kg	31,000/kg	-	15,318/kg
Length	38 mm	45 mm	19 mm	35 mm
Estimated Diameter	0.8 mm	0.8 mm	-	1 mm
Specific Gravity	0.91	0.91	2.55	7.93
Tensile Strength	570–660 MPa	750–850 MPa	570–660 MPa	1250–1350 MPa
Colour	Gray	Black	White	Silver
	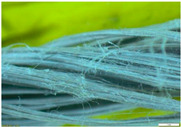	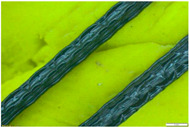	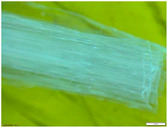	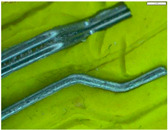

**Table 4 polymers-13-03008-t004:** Test programme for geopolymer mortar with short fibres.

Test	Fibre Type	Dosage		Number of Specimens
		% by Volume	Kg/m^3^	
Compression	Twisted PP fibre	0.1	0.91	3
	Straight PP fibre		0.91	
	Short glass fibre		2.55	
	Steel Fibre		7.93	
Splitting Tensile	Twisted PP fibre		0.91	
	Straight PP fibre		0.91	
	Short glass fibre		2.55	
	Steel		7.93	

**Table 5 polymers-13-03008-t005:** Test outline stage 5.

Test	Fibre Type	Dosage		Number of Specimens
		% by Volume	Kg/m^3^	
Compression	Straight PP fibre	0.1	0.91	3
		0.2	15.86	
		0.3	23.79	
		0.4	31.72	
		0.5	39.65	
Splitting Tensile	Straight PP fibre	0.1	0.91	
		0.2	15.86	
		0.3	23.79	
		0.4	31.72	
		0.5	39.65	

**Table 6 polymers-13-03008-t006:** Compressive strength of concrete with different fibre.

Fibre Type	Average *σ*_c_ (MPa)	Standard Deviation (MPa)	Coefficient of Variation (%)	Axial Toughness
				(MPa.mm)
Straight PP fibre	17.3	1.1	6.3	29.8
Twisted PP fibre	19.0	0.4	1.9	25.4
Short glass fibre	12.1	0.5	6.4	18.5
Steel Fibre	14.5	0.2	1.6	20.9
Control sample	11.6	1.2	9.9	12.6

**Table 7 polymers-13-03008-t007:** Splitting tensile strength (*σ*_t_) of geopolymer mortar with different fibre.

Fibre Type	Average *σ*_t_ (MPa)	Standard Deviation (MPa)	Coefficient of Variation (%)
Straight PP fibre	2.3	0.4	9.5
Twisted PP fibre	1.5	1.1	9.0
Short glass fibre	1.6	0.5	8.5
Steel Fibre	1.6	0.2	13.2
Control sample	0.6	1.2	17.1

**Table 8 polymers-13-03008-t008:** Compressive strength of geopolymer mortar with different dosages.

Fibre Dosage	Average *σ*_c_ (MPa)	Standard Deviation (MPa)	Coefficient of Variation (%)
0.91 kg/m^3^	19.1	0.15	0.78
15.86 kg/m^3^	24.4	0.20	0.81
23.79 kg/m^3^	25.0	0.49	1.96
31.72 kg/m^3^	31.1	1.45	4.66
39.65 kg/m^3^	32.2	0.15	0.46

**Table 9 polymers-13-03008-t009:** Tensile strength of geopolymer mortar with different dosages.

Fibre Type	Average *σ*_t_ (MPa)	Standard Deviation (MPa)	Coefficient of Variation (%)
0.91 kg/m^3^	2.3	0.02	0.86
15.86 kg/m^3^	2.9	0.2	6.89
23.79 kg/m^3^	2.8	0.2	7.14
31.72 kg/m^3^	4.1	0.2	4.87
39.65 kg/m^3^	5.4	0.4	7.4

**Table 10 polymers-13-03008-t010:** Comparison between the theoretical and experimental test results.

Fibre Dosage	Compressive Strength (MPa)	Equation (1) (MPa)	Error (%)	Tensile Strength (MPa)	Equation (2) (MPa)	Error (%)	Equation (3) (MPa)	Error (%)
0.0%	11.6	11.6	0%	0.7	2.1	425%	1.1	43%
0.1%	19.1	21.3	12%	2.3	2.7	101%	1.9	−15%
0.2%	24.4	24.9	2%	2.9	3.1	72%	2.5	−13%
0.3%	25.0	27.8	11%	2.8	3.1	88%	3.2	13%
0.4%	31.1	30.4	−2%	4.1	3.5	34%	4.1	0%
0.5%	32.2	32.7	2%	5.4	3.5	6%	5.2	−4%

## Data Availability

Not applicable.

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
