# Peer review of "Effect of Short Fibres in the Mechanical Properties of Geopolymer Mortar Containing Oil-Contaminated Sand"

_polymers, 2021, doi:10.3390/polym13173008_

Round 1
Reviewer 1 Report
This paper studied the mechanical properties of geopolymer mortar containing oil contaminated sand using short fibres. Some modifications are mentioned here to enhance the quality of the manuscript: 1. In page 1 line 19 and 22, is 6% by volume or by mass? 2. Most of the references in this paper are dated and it is suggested to add some papers on fiber-reinforced mortar in the last two years. For example, [1] Li L, Li Z, Cao M, et al. Nanoindentation and Porosity Fractal Dimension of Calcium Carbonate Whisker Reinforced Cement Paste After Elevated Temperatures (up to 900℃)[J]. Fractals. 2021, 29(2): 2140001. [2] Zeng D, Cao M, Ming X. Characterization of mechanical behavior and mechanism of hybrid fiber reinforced cementitious composites after exposure to high temperatures[J]. Materials and Structures. 2021, 54:26. 3. Will light crude oil contaminate concrete, causing environmental or durability problems? This should be discussed or explained in the paper. 4. Where is Figure 2? 5. The 50×100 mm size cylinders are used in this paper, but the fiber length in this paper is 19-45 mm. Generally, the fiber length should not exceed one third of the diameter of the specimen, so I think this size of the specimen is not appropriate in this paper. 6. Fibers enhance the toughness of mortar more than the compressive strength. Therefore, I suggest that compressive toughness be discussed in Section 3.1.1. 7. In my opinion, transverse deformation rather than longitudinal displacement should be discussed in splitting test in section 3.1.2. 8. In Figure 9, the stiffness of steel fiber is usually greater than that of PP fiber. Why is the strength of PP fiber reinforced concrete greater than that of steel fiber reinforced concrete? 9. In my opinion, in addition to fiber content, fiber characteristic parameters including aspect ratio are more meaningful. The influence of fiber characteristic parameters should be discussed in this paper and compared with studies in the literature. 10. The ordinate of Fig. 9 (a) should be the splitting tensile strength.Author Response
Please see the attached file

Reviewer 2 Report
In this manuscript the influence of short fibres on geopolymer mortar containing oil contaminated sand is investigated. Overall the paper is in good shape. The topic is interesting and the conclusion is solid. I have the following suggestions:
- Please insert one paragraph to illustrate the innovative part of this paper. There are available studies about related topics.
- Figure 7 is not clear. What is the size of the samples?
- Please clarify if the test properties are related to any engineering performance in field.
- What is the cost of the used fibres?
- More recent literature should be included.
- The modification mechanism of different fibres are same of different? Please clarify.
Reviewer 3 Report
- Although the article introduction implies the use of oil-contaminated sand, it focuses on short fibre geopolymers.
- The article compares the properties of 4 types of fibres rather than comparing with geopolymers containing non-contaminated sand.
- Comparison with short fibre reinforced geopolymers with non-contaminated sand mix for comparison is missing.
- In Table 3, it appears glass fibre properties are the same as virgin homopolymer polypropylene. What is the specific gravity of glass fibre? About 2.5? Please explain.
- σc is used for both compressive strength and tensile strength? Please distinguish between the two. Also in Table 6, are the twisted fibre and straight fibre properties interchanged (as compared to the figure)?
- Figure 9(a) shows nomenclature not used in the text. Please correct.
- How do authors gauge the quality of the fibre-matrix interface from Fig. 10 (as mentioned in the text).
- Figure captions appear to be too terse. Some more detail is needed.
- Why would the mechanical properties not improve with greater (.0.5%) fibre dosage?
- How do authors surmise that longitudinal fibres control internal cracking (l. 454-455) from Fig. 12.
- Discussion lacks a more rigorous treatment.
Round 2
Reviewer 2 Report
The paper has been obviously improved. It is now acceptable.
Reviewer 3 Report
Authors have performed a fair revision of the manuscript. However, a number of issues have to be addressed:
- In Table 3, is short glass fibre PP or glass? What do the authors mean by glass fibres?
- Table 4 shows the density of short glass fibre is still 0.91.
- The caption in Fig. 6 does fit the characteristics of a journal caption.
- What do lines 268-270 mean?
- Lines 579-582 appear to be in direct contrast to Lines 666-669.
- What do the arrows in Fig. 9 indicate? Please explain adequately in the caption.
- The manuscript should be thoroughly revised for a number of typos and grammar to substantially enhance it. E.g. compression and not comprission.
